# ArsenicNet: An efficient way of arsenic skin disease detection using enriched fusion Xception model

Md Humaion Kabir Mehedi[1], Kh. Fardin Zubair Nafis[1], Krity Haque Charu[1], Jia Uddin[2], Md Golam Rabiul Alam[1], M.F. Mridha[3]*

1 Department of Computer Science and Engineering, BRAC University, Dhaka, Bangladesh, 2 AI and Big Data Department, Endicott College, Woosong University, South Korea, 3 Department of Computer Science, American International University, Dhaka, Bangladesh

* firoz.mridha@aiub.edu

**Data availability statement:** We used the data from Mendeley Data link: https://data.mendeley.com/datasets/x4hgnjj5gv/2.

## Abstract

Arsenic contamination of drinking water is a significant health risk. Countries such as Bangladesh's rural areas and regions are in the red alert zone because groundwater is the only primary source of drinking. Early detection of arsenic disease is critical for mitigating long-term health issues. However, these approaches are not widely accepted. In this study, we proposed a fusion approach for the detection of arsenic skin disease. The proposed model is a combination of the Xception model with the Inception module in a deep learning architecture named "ArsenicNet." The model was trained and tested on a publicly available image dataset named "ArsenicSkinImageBD" which contains only 1287 samples and is based on Bangladeshi people. The proposed model achieved the best accuracy through proper experimentation compared to several state-of-the-art deep learning models, including InceptionV3, VGG19, EfficientNetV2B0, ResNet152V2, ViT, and Xception. The proposed model achieved an accuracy of 97.69% and an F1 score of 97.63%, demonstrating superior performance. This research indicates that our proposed model can detect complex patterns in which arsenic skin disease is present, leading to a superior detection performance. Moreover, data augmentation techniques and earlystoping function were used to prevent models overfitting. This study highlights the potential of sophisticated deep learning methodologies to enhance the accuracy of arsenic detection and prevent premature interventions in the diagnosis of arsenic-related illnesses in people. This research contributes to ongoing efforts to develop robust and scalable solutions to monitor and manage arsenic contamination-related health issues.

## 1 Introduction

Arsenic is one of the most dangerous diseases found in the environment in various forms, especially in the outermost layer of a terrestrial plane. It is a toxic metalloid that is frequently found in water, and its after-effects can cause dangerous health issues such as arsenicosis or arsenic poisoning. Since the early 1990s, arsenic contamination has been considered a serious public health problem [1]. From the beginning, scientists found that drinking

**Funding:** The author(s) received no specific funding for this work.

**Competing interests:** The authors have declared that no competing interests exist.

groundwater was one of the main reasons for spreading arsenic. Nevertheless, the problem created a dark line for researchers when millions of tube wells, which were leveled as safe drinking water, were contaminated with arsenic at levels far exceeding the WHO's (World Health Organization) suggested limit of 10 parts per billion (ppb).

It is important to detect arsenic in drinking water because its long-term effects are terrible and pervasive. Arsenic can cause threatened health problems such as skin lesions, cardiovascular diseases, diabetes, and different forms of cancer, mainly skin, lung, and bladder cancers. In addition, researchers have found that arsenic can be associated with developmental issues in children, such as memory loss and unfortunate pregnancy outcomes, including increased risks of stillbirth and infant mortality. Therefore, detecting arsenic in water is crucial for public health researchers.

In Bangladesh, the aftermath of arsenic contamination is boundless and affects millions of people throughout the country. According to researchers, approximately 20–40 million people in our country are being attacked by drinking arsenic-contaminated water, creating the most black-marked poisoning incidents in history. The presence of arsenic in drinking water varies from region to region. However, most occurrences have been found in our country's southern and southwestern parts, where arsenic levels are dangerously high [2]. Rural people are mostly affected by arsenic because there are very few alternative drinking water options.

Researchers have revealed that in any gender or age, people can be attacked by arsenic, although particular groups are more at risk owing to their higher water consumption scales or long-term exposure periods [3]. Children are among them at an increased risk of developmental problems, while women can experience some symptoms during pregnancy. Arsenic can also cause serious economic problems. This can affect the livelihoods of those affected by arsenic. This can increase healthcare costs for the suffering families.

Our country's responses to the control of arsenic contamination were very significant. Researchers have shifted from purifying contaminated tubewell water to adopting more advanced technology for modern solutions. Initially, people followed traditional methods such as sedimentation and filtration to decrease arsenic levels in water [4]. However, these methods are currently too few to consider when providing safe water at the scale. Recently, many new advanced solutions have been proposed to control the contamination of arsenic. For example, installing community-based arsenic removal plants, creating awareness of rainwater, harvesting, and setting up arsenic filters. Moreover, some machine learning models have also been introduced to detect the symptoms of arsenic disease so that people can receive early treatment if they are affected.

Based on previous work, this research proposes a novel approach to detecting arsenic skin disease early and effectively. To accomplish this, a fusion deep learning model named "ArsenicNet," is introduced, which is a combined Xception model with the Inception module. Furthermore, the "ArsenicSkinImageBD" dataset is used to train and test the model. In addition, the dataset was preprocessed and augmented prior to the training. Moreover, other state-of-the-art models were trained and tested on the given dataset. Finally, the ArsenicNet model achieved better performance than the others.

Main findings of this research:

- We are the first to work on image-based arsenic skin disease detection in South Asia. The data set has two classes: one is infected and the other is normal. All the previous studies are based on tabular data.

- We applied different preprocessing techniques such as horizontal and vertical flips and shifts. In addition, a Gaussian filter was used to sharpen the image and achieve better performance.
- We have proposed a fusion model, which is a combined Xception model with an Inception module containing multi-scale feature extraction and can predict disease effectively.
- We conducted extensive experiments and comparative analysis with other state-of-the-art models using different evaluation methods such as accuracy, precision, recall, F1 score, ROC, and confusion matrix.

This research paper introduces a new methodology for the classification of arsenic skin disease, aiming to identify the crucial factors that impact the decision-making process. The remainder of this study is structured as follows:

- Section 2, Related Works: provides an overview of the recent findings regarding arsenic detection. Here we have discussed some novel existing approaches that help us to research more deeply in our project.
- Section 3, Methodology: delves into our proposed model's and other existing model's architectures with fine-tune tables. Besides, the data processing part is also included in this section.
- In Section 4, Experiment setup and Result Analysis:the results obtained from the classification process are highlighted and deeply researched. Experimental setup is also discussed elaborately in this segment. In addition, some graphs and confusion matrices have been added with detailed explanations of our result.
- Finally, Section 5, Conclusion: concludes the research paper by summarizing the key findings and their application derived from the research. This research highlights a novel approach to detecting arsenic skin disease in the human body. The remarkable performance of the proposed model in detecting complex patterns in arsenic-contaminated images can prevent several types of health issues.

## 2 Related works

There are several studies in which we can obtain information about arsenic disease detection in our skin using deep learning. The authors tried to get information for measuring arsenic in groundwater and wanted to draw a concern regarding the negative effects of arsenic pollution on the environment and public health using several techniques and evaluating multiple factors [5]. Moreover, classifying arsenic in groundwater in Jharkhand, India, is the main challenge for researchers [6]. Here, the authors focused on how arsenic poses severe health issues. Taking some important features such as land use and soil composition, the authors studied various models to obtain the best performance and accuracy, where a random forest was picked for the desired result. This paper used boosted regression trees (BRT) and random forests (RF) as advanced machine learning techniques to detect arsenic in groundwater. With help from key factors like topography, lithology, erosion, hydrology, and human effects, the author distinguished between contaminated arsenic in water and normal water, which was the main outcome for this published work [7]. The Receiver Operating Characteristics (ROC) curve is used here to measure the model's performance. This methodology helps those areas where arsenic is present in groundwater, focusing on important variables like proximity to the living area and precipitation.

This research used the common algorithm of machine learning and random forest for classifying and regression tasks in detecting arsenic in drinking water in west-central

New Jersey [8]. Focusing on relationships between many environmental and anthropogenic factors like land use, bedrock type, soil type, drainage class, proximity to risky sites, abandoned mines, and orchards in drinking water, this model can detect arsenic with an accuracy of 55% in regression and 66% for classification. Offering a basis for developing focused outreach and intervention plans, these factors help the author to identify arsenic in water. Removing the public's lack of awareness and seriousness regarding arsenic was this study's main goal. Most of us know that arsenic can be found in water; however, in this discussion paper, the authors have attempted to detect arsenic in rice consumption in regions where arsenic contamination is prevalent [9]. The authors detected bioavailable arsenic in paddy soil using validated decision tree (DT) and logistic regression (LR) models, where the DT model gave better accuracy than LR. Using different variables, this paper provides information that a guideline value of 14 mg kg$^{-1}$ for total arsenic in paddy soil can be found. Besides, the authors share that the LR model can identify significant factors affecting arsenic in rice grains. Overall, this paper gives us a general overview of detecting arsenic in soil, which helps to create awareness among local people. Though most of the paper uses different approaches for machine learning, in this paper, the author disclosed a novel approach called SHapley Additive exPlanation (SHAP), which can measure the performance of different machine learning models [10]. Explaining Computing the contribution of every factor of all models, this model is based on the SHapley value from cooperative game theory. SHAP can help to learn which model can outperform in detecting arsenic in water. Besides, this model uses extreme gradient boosting (XGB), which performs well between all models regarding matrix measures.

Furthermore, the paper addresses the critical issue of groundwater arsenic contamination, affecting 2.5 billion people globally who rely on groundwater for drinking and irrigation. The World Health Organization (WHO) suggests a 10 μg/L guideline for arsenic concentration in groundwater [11]. Prolonged exposure to arsenic-contaminated water poses various health risks. The study introduces a geospatial-based machine learning method to classify arsenic levels into high (1) or low (0) using environmental factors. Groundwater samples from Varanasi district, India, along the river Ganga, were analyzed, and various machine learning models were compared. The deep neural network (DNN) model exhibited superior performance with 92.30% accuracy, 100% sensitivity, and 75% specificity. Policymakers can leverage the DNN model's accuracy to identify at-risk individuals and develop targeted mitigation strategies using spatial maps. Detecting arsenic and manganese in glacial aquifers is a novel finding of this paper [12]. The boosted regression tree (BRT) model is used in this paper with Bernoulli distribution to predict it more accurately. After doing some effective tuning and testing, this model can achieve the accuracy rate for arsenic at 90% and manganese at 83%. This model can also help find arsenic in groundwater. In Ghana, several people were attacked by arsenic disease. This paper is dedicated to that region to detect arsenic using machine learning. Here, authors tried to use some well-known approaches like decision trees (DT), multivariate adaptive regression splines (MARS), multilayer perceptron neural networks (MLP), random forests, and combining with extreme gradient boosting (XGB), light gradient boosting (LGB), and generalized regression neural networks (GRNN). Including the coefficient of determination (R2), Nash-Sutcliffe efficiency (NSE), and mean squared error (MSE), all models perform well in detecting arsenic in water, which can be a threat to drinking water [12]. By combining a Fruit Fly Optimization Algorithm (FOA) with a Support Vector Machine (SVM), this study introduces a groundbreaking machine learning technique that effectively classifies skin cancer from dermoscopic images [13].

The main objective of the FOA-SVM method is to enhance the accuracy of diagnosis and address the limitations associated with traditional cancer detection methods, thereby assisting

physicians in making informed decisions. The method's accuracy in classifying skin lesions demonstrated its potential as a useful tool for clinical diagnosis. Future research efforts will focus on improving the method to correct shortcomings and expand its ability to diagnose diseases. Moreover, to detect skin cancer, the study introduces a computer-assisted analysis technique that merges the contour transform (CT) and local binary pattern (LBP) [14]. These methods are enhanced using Particle Swarm Optimization (PSO). It helps to lessen computing expenses and enhances extracting characteristics for identifying skin lesions. According to the study, SVM exhibits the lowest time complexity compared to Random Forest and Neural Networks. The approach achieves an accuracy level of 86.9%, indicating a promising direction for reliable and precise medical diagnostic tools. This study presents an innovative near-infrared (NIR) spectroscopic method called NIR-SC-UFES, designed specifically for collecting spectral data about prevalent skin lesions [11]. This paper introduces a novel approach to detecting skin cancer using a machine learning algorithm [15]. Several algorithms are used in this paper, such as LightGBM, CatBoost, XGBoost, 1D-CNN, and so on. After completing feature extraction and preprocessing, LightGBM demonstrated the most optimal outcomes. The study assists in identifying skin cancer by presenting a new dataset and demonstrating the capabilities of NIR spectroscopy in medical diagnostics. The study suggests a new approach for identifying skin cancer using a combination of classifiers and advanced techniques to extract detailed features from real-time data [16]. The study utilizes an enhanced harmony search technique, a sand cat swarm optimization approach, and ResNet50 to extract features and reduce dimensions. We conducted several experiments with different classifiers like Naive Bayes and SVM using the Kaggle and ISIC 2019 datasets. The findings showed a significant improvement in the precision of forecasts. This method can improve the results for patients by helping to promptly detect skin cancer. Besides, the research presents an innovative approach to unveiling facial skin issues, empowering individuals to self-diagnose their skin conditions without consulting an expert in person. To detect skin diseases, this paper has followed some novel approaches. The InceptionV3 model attained an accuracy rate of 94%, in contrast to the VGG16 model, which only managed to reach 74% [17]. In the forthcoming times, scientists will primarily focus on advancing mobile-compatible models, incorporating various modes of data and extending the application of deep learning techniques to a broader array of skin ailments. The study showcases the possibilities of using automated methods in diagnosing dermatological conditions. The paper suggests a CAD system that uses DCNN techniques. This system categorizes dermoscopy images as melanoma, benign, malignant, or human against the machine (HAM) categories [18]. The study attains a 97.25% accuracy rate and recommends optimizing hyperparameters and experimenting with various pre-trained CNN models for better future results. This technology aims to automatically detect skin cancer, speed up diagnosis, and potentially save lives. The novel technique called Limited Samples Network (LSNet) is presented in this article to address the challenges of using deep learning for skin lesion image interpretation due to a lack of training data. It, however, introduces a revolutionary technique called Limited Samples Network (LSNet), which finds and uses difficult samples to boost learning [19]. An autoencoder that learns pseudo-inverse, as well as patch-based structured input, is used by this approach for position-sensitive loss determination. This method is an improvement over other methods in terms of solving transfer learning problems on several ISIC datasets without additional data, and it outperforms traditional pre-trained DCNNs regarding performance.

In their innovative research, they proposed LeaNet, a unique U-shaped network for lightweight and high-performance skin cancer picture segmentation [20]. This targets low powered medical gadgets that use multiple focus blocks to improve the local and global feature learning necessary for accurate lesion boundary delineation. LeaNet outperformed

ResUNet regarding accuracy, sensitivity, and specificity while reducing the parameter count and computational complexity on the ISIC2017 and ISIC2018 datasets; thus, it is state-of-the-art. In addition, lighter models than MALUNet are also beaten by it. This study aimed to enhance skin cancer diagnostics by fusing dermatological knowledge with computational methods [21]. It provides a method for extracting features from images that combines PCA and autoencoders, which aim to retain only important information. These attributes are then used in machine learning models, such as XGBoost and LSTM, to enhance the diagnosis procedure. The results confirm that these models can accurately detect skin cancer and help doctors during clinical procedures and patients' welfare. This study presents NIR-SC-UFES, a new method for gathering NIR spectral data for ordinary skin lesions [15]. Research shows how to detect skin cancer through machine learning algorithms such as XGBoost, CatBoost, LightGBM, and 1D-CNN, which apply data augmentation methods due to class imbalances such as SMOTE and GAN. After feature extraction and preprocessing, the LightGBM exhibited the best performance. This research helps identify skin cancer by providing a new dataset and demonstrating the potential of NIR spectroscopy in medical diagnostics.

Table 1 shows the summary of the previous state-of-the-art works in this domain.

## 3 Methodology

In this study, we have followed several key steps. Fig 1 shows the steps involved in the proposed methodology. First, we have collected a dataset for the work. The data were then preprocessed. Next, we applied the proposed fusion deep learning model. Finally, we compare our proposed model with other state-of-the-art models based on evaluation metrics.

**Table 1. Summary table of previous state-of-the-art works.**

| Ref. | Model | Performance | Dataset Type | Dataset |
|---|---|---|---|---|
| [5] | Optimized Forest | Accuracy 80.64% | Tabular | Groundwater samples from Varanasi, India |
| [7] | Random Forest | AUC 85% | Tabular | Arsenic concentrations in southeastern Michigan |
| [8] | Random Forest | Accuracy 66% | Tabular | Drinking water wells in west-central New Jersey |
| [10] | Eextreme Gradient Boosting | Accuracy 86% | Tabular | Ghana Water Samples |
| [11] | Deep Neural Network (DNN) | Accuracy 92.3% | Tabular | River Ganga's banks of Varanasi Samples |
| [12] | Boosted Regression Trees | Accuracy 90% | Tabular | U.S. Glacial Aquifer System |
| [13] | Hybrid Framework | Accuracy 87.02% | Image | HAM10000, ISIC2018, ISIC2019 |
| Our | Proposed ArsenicNet | Accuracy 97.69% | Image | ArsenicSkinImageBD |

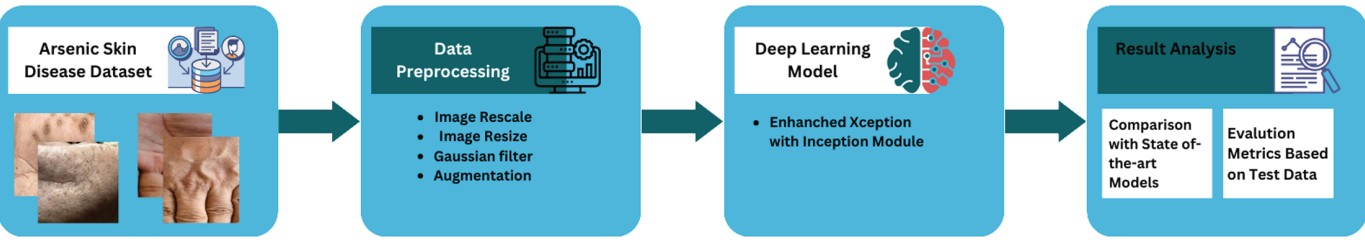

**Fig 1. Workflow of the proposed system.** First, data collection, then data preprocessing, including augmentation and Gaussian filter, followed that applying deep learning models, and finally, result analysis.

### 3.1 Dataset description

The dataset was collected from Mendeley data. The name of this dataset is "ArsenicSkinImageBD" [22]. The dataset sample has been collected from four different villages in Bangladesh.

The villages are: Betbaria, Ward:08, Union: Balidanga; Balubagan, Ward:08, Union: Maharajpur; Dole para, Ward:09, Union: Maharajapur; and Ramchandrapur hat, Ward:08, Union: Dohilpara, Sadar, and Chapainawabganj. Sample pictures were captured using four different smartphones. It contains two unique class, one is infected and another one is not-infected or normal. It has a total of 1287 samples altogether; the infected class has 741 and the not-infected has 546 samples. Some sample images of the dataset are shown in Fig 2.

### 3.2 Data preprocessing

The images we obtained from the dataset were different in size, so we resized them to $224 \times 224$. The images were also rescaled to $[0-1]$. To increase the effectiveness of our proposed method, we conducted data augmentation and a Gaussian filter for noise reduction.

1. Data Augmentation

   We first applied data augmentation techniques, including horizontal flipping true, width and height shift of 0.10 each, but excluding vertical flips [23]. These additions make it appear that the subjects are in different places and at different angles in the pictures. This makes it easier for the model to find patterns regardless of their angle or alignment in the input data. The defined values are listed in Table 2.

2. Gaussian Blur

   A Gaussian filter, which is also sometimes called a "Gaussian blur," is an important tool in image processing for smoothing out pictures so that they look less sharp and have less noise [24]. In data processing, blurring is a strategy that can be used to smooth out the values gathered in the dataset. The data were reprocessed using the Gaussian blur technique to enhance accuracy.

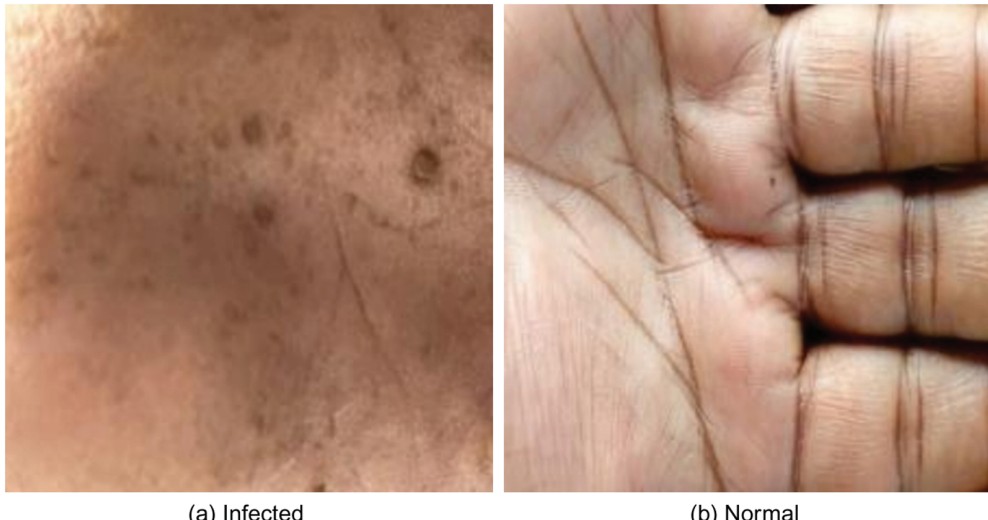

(a) Infected                    (b) Normal

**Fig 2. Samples of arsenic infected and normal skin images.**

**Table 2. Data augmentation and resize values for this study.**

| Name of the Techniques | Range |
|---|---|
| Resize of images | $224 \times 224 \times 3$ |
| Horizontal flip | True |
| Vertical flip | False |
| Rescale | 1./255 |
| Width shift range | 0.1 |
| Height shift range | 0.1 |

### 3.3 Data split

We then tried different data split values, such as 70:15:15 and 80:10:10. However, 80:10:10 split set yielded good results for our models. Here, 80% for training set, 10% for validation set and rest of the 10% for test set.

### 3.4 Proposed model (Xception + inception module)

The inception module enhances the model's efficiency and efficacy via Xception, which uses depthwise separable convolutions to extract strong spatial features uniformly across the feature map. In contrast, a multi-scale processing module such as Inception helps the model learn features of different sizes and complexity. The fine-grained features of Xception are enhanced by the Inception module, making the model more adaptable to different datasets. The modular design of Inception and Xception's computational efficiency makes the architecture lightweight. The trainable parameters for Xception were 51,512,578, whereas the proposed combination had 814,962. The fixed Xception architecture can be flexible by changing the filter sizes or branch configurations in the Inception module to suit the dataset. Medical images, such as skin disease images, benefit from this combination because the Xception backbone extracts strong features, and the Inception module captures context and scale-specific information.

The proposed model combines the pre-trained Xception and an Inception module. First, we used preprocessed data as input for the model. Then, we used Xception as the backbone model by making "include_top= False" and "weights='imagenet'." Next, a single Inception module uses different filters for distinct convolution operations. After that, global average pooling was used to reduce the model's complexity and the dimension of the features. Moreover, a dense layer is used with 1024 neurons and the "relu" activation function. Finally, the model concludes with a dense output layer with 2 neurons and a "sigmoid" activation function. Fig 3 represents the proposed model of this paper.

$$\mathbf{F}_X = \text{Xception}(\mathbf{I}), \quad \mathbf{F}_X \in \mathbb{R}^{H' \times W' \times C'} \tag{1}$$

where $H'$, $W'$, and $C'$ are the height, width, and number of channels in the extracted feature maps, respectively. The backbone of the Xception model is in Eq. 1.

The Inception module computes multi-scale features through multiple branches with different filters and kernel sizes in Eqs. 2–6:

$$\mathbf{F}_{1\times1} = \mathbf{K}_{1\times1} * \mathbf{F}_X \tag{2}$$

where

- $\mathbf{F}_{1\times1}$: Output feature maps from the $1 \times 1$ convolution branch.

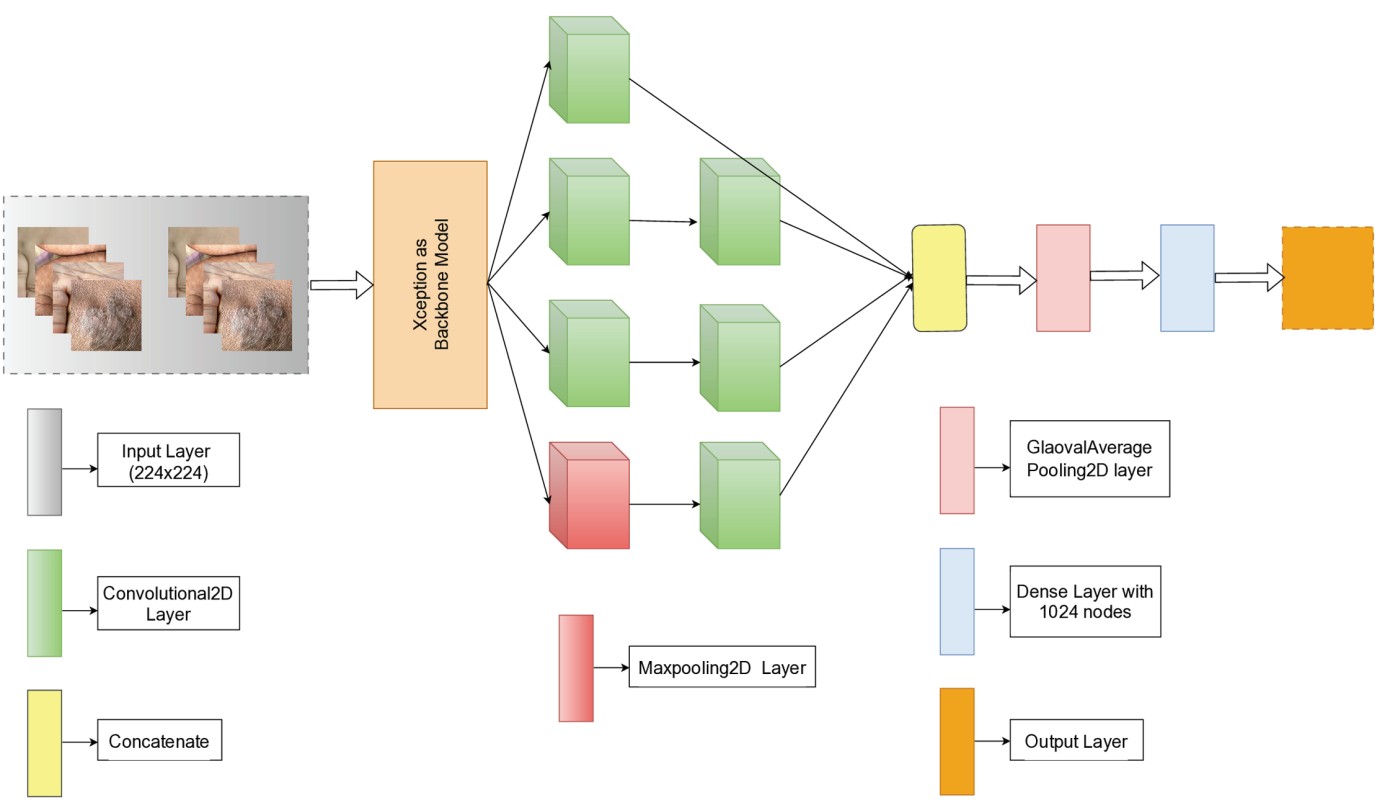

**Fig 3. Architecture of the proposed model. Xception as the backbone model and added the Inception module with it.**

- $\mathbf{K}_{1\times1}$: Weights of the $1 \times 1$ convolution kernel.
- $\mathbf{F}_{X}$: Input feature maps from the Xception backbone.

$$\mathbf{F}_{3\times3} = \mathbf{K}_{3\times3} * \left( \mathbf{K}_{\text{reduce3}} * \mathbf{F}_{X} \right) \tag{3}$$

where

- $\mathbf{F}_{3\times3}$: Output feature maps from the $3 \times 3$ convolution branch.
- $\mathbf{K}_{\text{reduce3}}$: Weights of the $1 \times 1$ convolution kernel for dimensionality reduction.
- $\mathbf{K}_{3\times3}$: Weights of the $3 \times 3$ convolution kernel.
- $\mathbf{F}_{X}$: Input feature maps from the Xception backbone.

$$\mathbf{F}_{5\times5} = \mathbf{K}_{5\times5} * \left( \mathbf{K}_{\text{reduce5}} * \mathbf{F}_{X} \right) \tag{4}$$

where

- $\mathbf{F}_{5\times5}$: Output feature maps from the $5 \times 5$ convolution branch.
- $\mathbf{K}_{\text{reduce5}}$: Weights of the $1 \times 1$ convolution kernel for dimensionality reduction.
- $\mathbf{K}_{5\times5}$: Weights of the $5 \times 5$ convolution kernel.
- $\mathbf{F}_{X}$: Input feature maps from the Xception backbone.

$$\mathbf{F}_{\text{pool}} = \text{MaxPool}(\mathbf{F}_{X}) \tag{5}$$

where

- $\mathbf{F}_{\text{pool}}$: Feature maps produced by the max pooling operation.
- $\mathbf{F}_{\text{X}}$: Input feature maps from the Xception backbone.

$$\mathbf{F}_{\text{pool1x1}} = \mathbf{K}_{\text{pool1x1}} * \mathbf{F}_{\text{pool}} \tag{6}$$

where

- $\mathbf{F}_{\text{pool1x1}}$: Output feature maps from the $1 \times 1$ convolution applied to the pooled feature maps.
- $\mathbf{K}_{\text{pool1x1}}$: Weights of the $1 \times 1$ convolution kernel.
- $\mathbf{F}_{\text{pool}}$: Input feature maps produced by the pooling operation.

The concatenated output of the proposed Inception module is in Eq. 7:

$$\mathbf{F}_{\text{Inc}} = \text{Concat}\left(\mathbf{F}_{1 \times 1}, \mathbf{F}_{3 \times 3}, \mathbf{F}_{5 \times 5}, \mathbf{F}_{\text{pool1x1}}\right) \tag{7}$$

The overall equation of the proposed model can be represented as in Eq. 8:

$$\mathbf{Y}_{\text{pred}} = \text{Sigmoid}\left(\mathbf{W}_{\text{Out}} \cdot \text{ReLU}\left(\mathbf{W}_{\text{Dense}} \cdot \text{GlobalAvgPool}\left(\text{Inc}(\text{Xception}(\mathbf{I}))\right) + \mathbf{b}_{\text{Dense}}\right) + \mathbf{b}_{\text{Out}}\right) \tag{8}$$

where

- $\mathbf{Y}_{\text{pred}}$: Final output predictions of the proposed model.
- $\mathbf{I}$: Input image of dimensions $224 \times 224 \times 3$.
- $\text{Xception}(\mathbf{I})$: Feature maps extracted by the pre-trained Xception backbone.
- Inc: The Inception module applied to the feature maps.
- GlobalAvgPool: Global average pooling layer for extarct global features.
- $\mathbf{W}_{\text{Dense}}, \mathbf{b}_{\text{Dense}}$: Weights and biases of the dense layer.
- $\mathbf{W}_{\text{Out}}, \mathbf{b}_{\text{Out}}$: Weights and biases of the output layer.

The inception module enhances the model's efficiency and efficacy via feature extraction. This model employs multi-scale feature extraction to process the input concurrently at several sizes. The Inception module employs concatenated outputs from the convolutional and pooling operations to generate a multi-dimensional feature map. It allows the model to capture various features from a single input and effectively integrate local and global information, enhancing its predictive accuracy. The module's capacity to collect characteristics at many scales boosts its proficiency in recognizing intricate patterns and structures, enhancing performance in tasks such as image classification. From Fig 4, we can see the block diagram of the inception module in detail. The output of the Xception (backbone) model was used as the module's input. Then, four parallel operations are executed, where three of them are convolution operations and 1 maxpool operation. Subsequently, a second layer of convolution operations is executed. Each operation uses a different kernel and a different number of filters. Finally, a concatenate layer combines all the output from the previous operation and forwards it to the next layer.

Different hyperparameters have been tested for this experiment with the proposed model to reach the peak position. Table 3, provides the tested and optimized value for the proposed model.

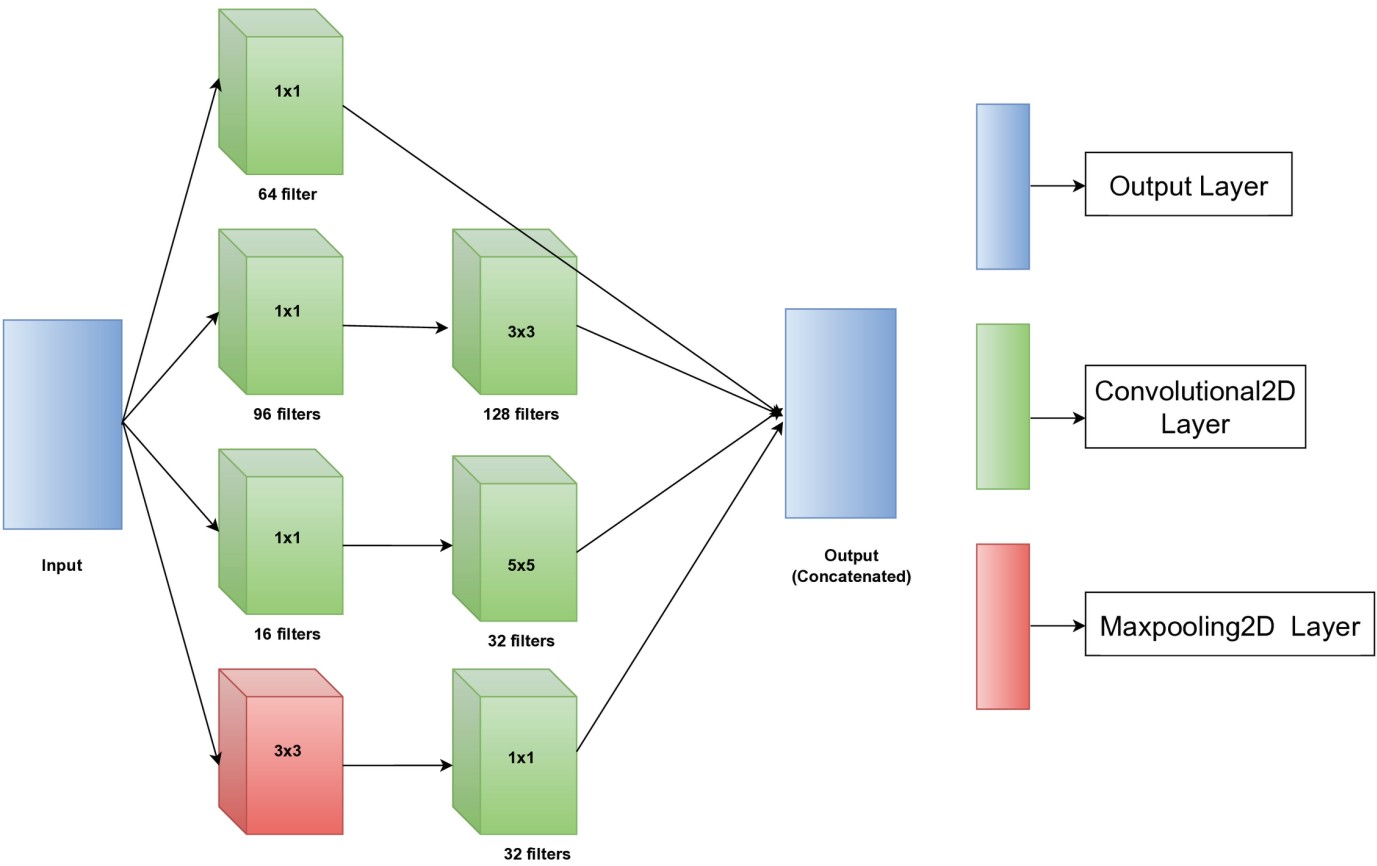

**Fig 4. Block diagram of the inception module with their kernel size and number of filters for each layer.**

Table 3. Hyperparameters for proposed model.

| Hyperparameters | Values | Optimized value |
|---|---|---|
| Image size | 128 × 128, 224 × 224 | 224 × 224 |
| Epochs | 100 | 20 |
| Learning rate | 1e–3, 1e–4, 1e–5 | 1e–4 |
| Batch size | 16, 20, 24 | 20 |
| Optimizer | Adam, SGD | Adam |
| Training-validation-test split | 70:15:15, 80:10:10 | 80:10:10 |

Table 3 represents the hyperparameters for the proposed model, presenting initial values and optimized values. The optimized hyperparameters include an image size of 224 × 224, 20 epochs, a learning rate of 1e–4, a batch size of 20, and the Adam optimizer. The data split ratio is 80-10-10.

## 4 Experiment setup and result analysis

### 4.1 Experiment setup

The authors adopted the following configurations to conduct the investigation. Table 4 provides a visual representation of the specific versions of the programming languages and

**Table 4. Elaboration of the setup for the research.**

| System | Configuration | Properties |
|---|---|---|
| Operating system | Windows | 11 |
| Programming Language | Python | 3.9.19 |
| Hardware | CPU | Ryzen 5 3600x |
| | RAM | 24 GB |
| | GPU | Nvidia RTX 4070 12 GB |
| Software | Framework | TensorFlow v2.11 |
| | CUDA toolkit | 11.2 |
| | DNN runtime library | cuDNN v8.7 |
| | IDE | Anaconda Jupyter notebook |

libraries used. In addition, the Table provides further information on the hardware and software requirements.

## 4.2 Results analysis

For this experiment, we applied 7 deep learning models; among them 6 are state-of-the-art models and our proposed fusion Xception model. All models were fine-tuned to obtain the best results. Despite training in the same environment, the models achieved optimal results on different hyperparameters. Different evaluation metrics [25] were used to compare the models.

The different deep learning models, such as InceptionV3 [26], VGG19 [27], EfficientNetV2B0 [28], ResNet152V2 [29], and Xception [30], have shown varying distinguishing characteristics in the classification of arsenic diseases. For all the models described so far, evident variations have been recorded while making true positive, true negative, false positive, and false negative predictions. Confusion matrices of all the models shown in Fig 5–7. An

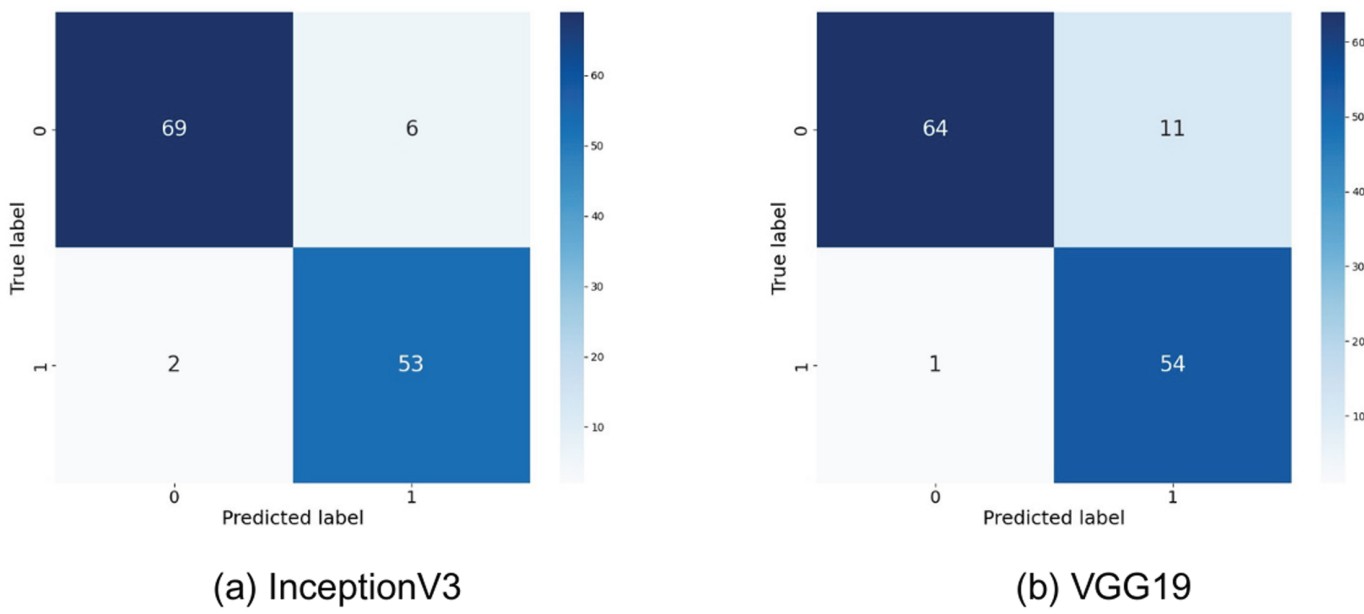

**Fig 5. Confusion matrices for InceptionV3 and VGG19.** Here, 0 = infected, 1 = normal.

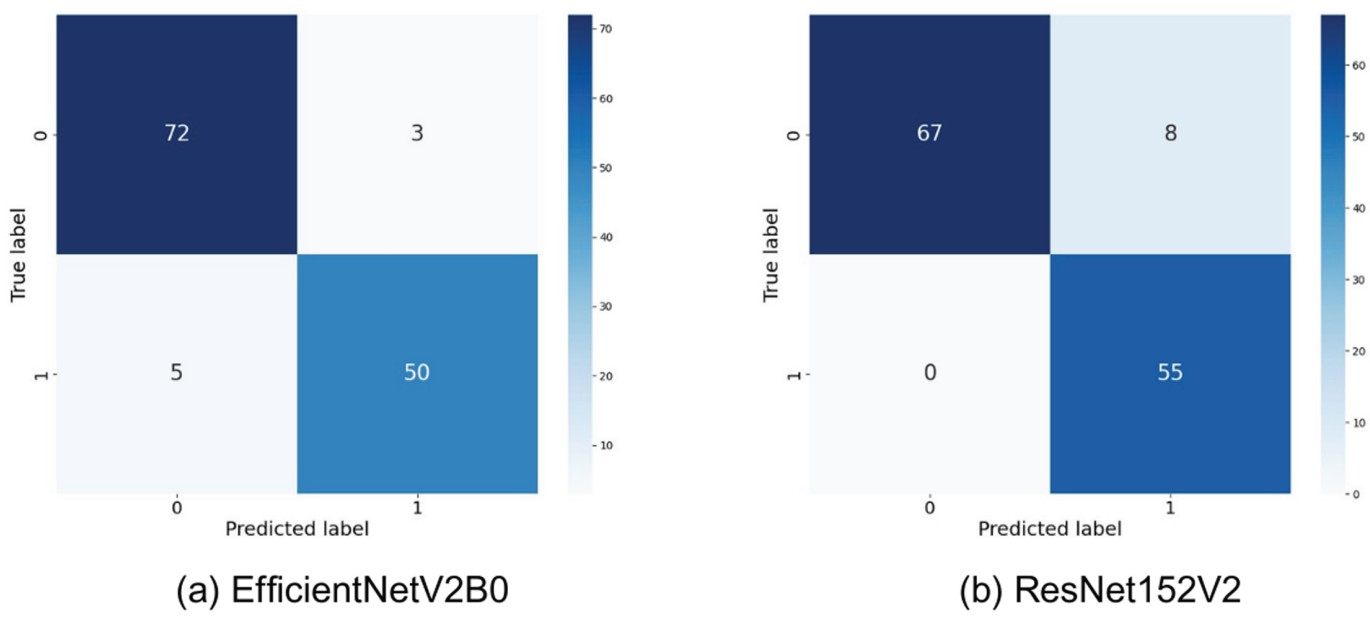

(a) EfficientNetV2B0

(b) ResNet152V2

**Fig 6. Confusion matrices for EfficientNetV2B0 and ResNet152V2.** Here, 0 = infected, 1 = normal.

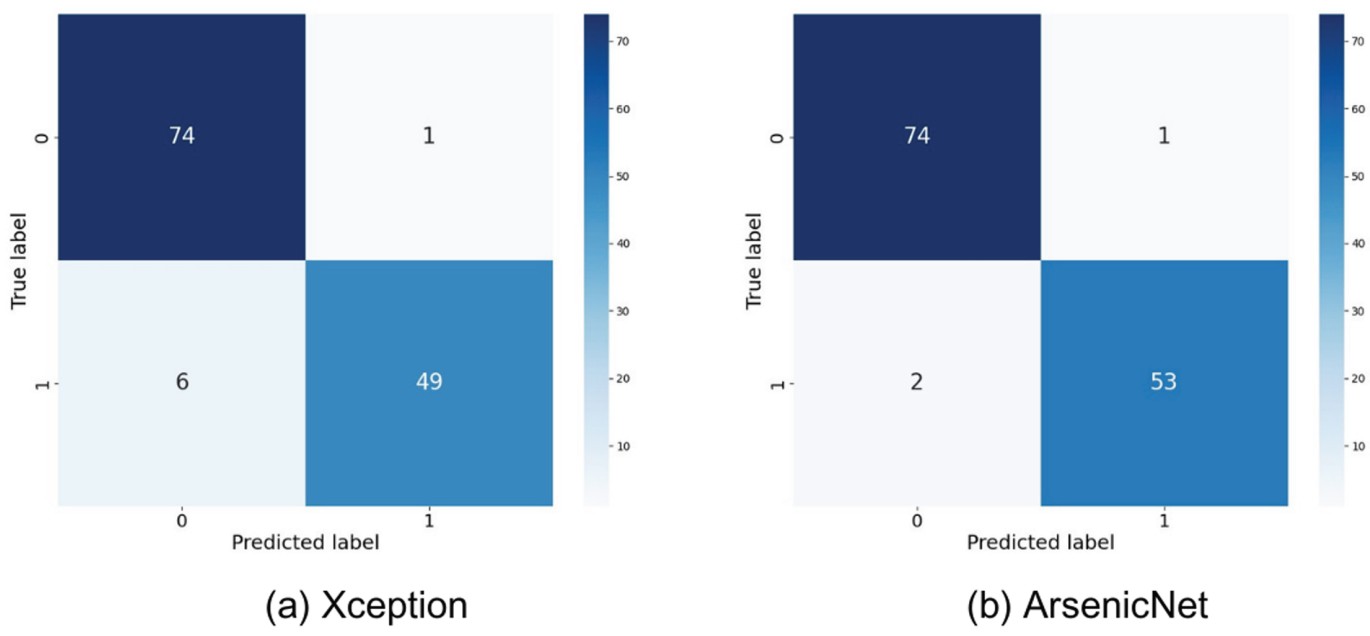

(a) Xception

(b) ArsenicNet

**Fig 7. Confusion matrices for Xception and Proposed model.** Here, 0 = infected, 1 = normal.

example is the highly demonstrated accuracy in the Xception model, with 74 true negatives and 49 true positives. Out of these, it classifies only 1 negative case as positive. It misses 6 positive cases, indicating a well-balanced performance with fewer false positives than VGG19 and a decent number of correct positive predictions, although slightly behind ResNet152V2, which had 0 false negatives. The other two models perform similarly, as InceptionV3 exhibits

a slightly higher number of false positives and negatives, while EfficientNetV2B0 exhibits the highest number of correct negative predictions. Although VGG19 worked well for the positives, it has quite a high false positive rate. ResNet152V2 is good in the true positives with no false negatives and has a moderate value of false positives. In general, Xception and ResNet152V2 are powerful in reducing false positives, whereas the best model title depends on whether the focus is on minimizing false positives or false negatives.

The confusion matrix of our proposed model strongly outperforms these pre-trained models in Fig 7. Our proposed model correctly identifies 74 cases of negative and 53 cases of positive. It resembles Xception in true negative cases but presents fewer false negatives. More precisely, the model makes only 1 incorrect prediction of a false positive and 2 incorrect predictions of a false negative. Compared to Xceptions 6 false negatives, this performance also overcomes that boundary. Our model had, when compared with others, a significantly lower number of false negatives, particularly against models like InceptionV3 and EfficientNetV2B0. Although ResNet152V2 had no false negatives, it experienced an increased incidence of false positives, making it more highly balanced. This lesser number of false positives and false negatives implies the reliability of the proposed approach, which will result in fewer missed detections and false alarms. This makes the proposed model superior in reducing the misclassification risk, which matters most for medical diagnosis due to the consequences of the false positive and false negative classes. The corresponding accuracies and the lower error rates the model has maintained infer a stronger and sufficiently suitable model for tasks related to tasks with higher classification, such as the detection of cancer.

This paper compares the performance of different models, including InceptionV3, VGG19, EfficientNetV2B0, ResNet152V2, Xception, and the proposed model, to classify the arsenic diseases. The ROC curve plots the true positive rate against the false positive rate and provides insight into how well each model can differentiate between the two classes.

InceptionV3, VGG19, and EfficientNetV2B0 showed outstanding performance with an AUC close to 0.99, which means that these models are relatively good at differentiating between positive and negative cases, as shown in Figs 8 and 9. ResNet152V2 is also very promising, as its ROC curve hugs the top left corner with high conformity, which means a high capacity of network coverage to minimize false positives and maximize true positives.

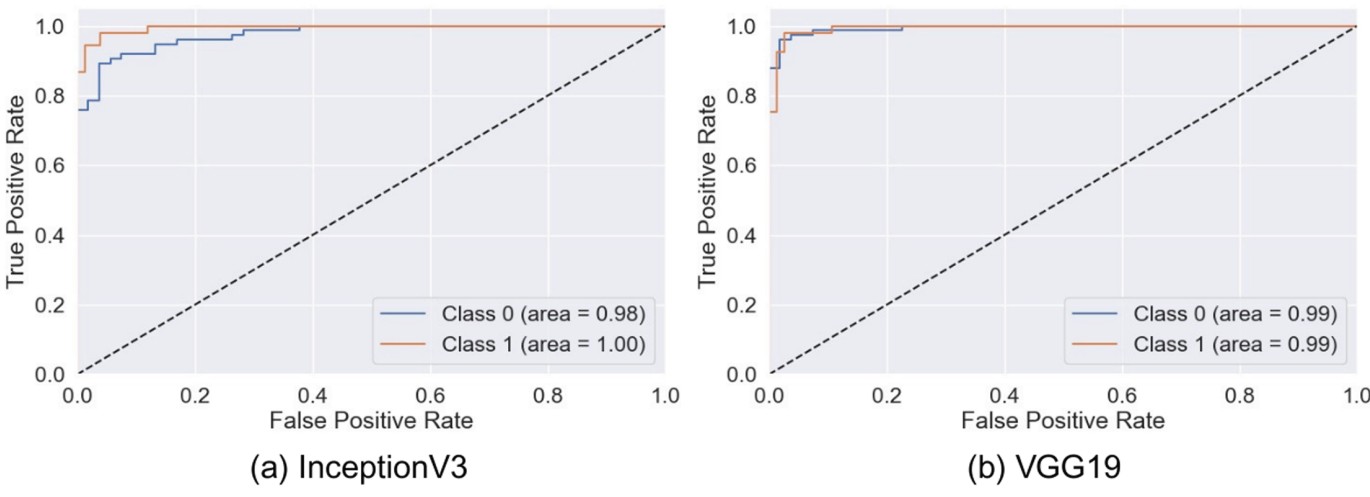

**Fig 8. ROC curves for InceptionV3 and VGG19.** Here, 0 = infected, 1 = normal.

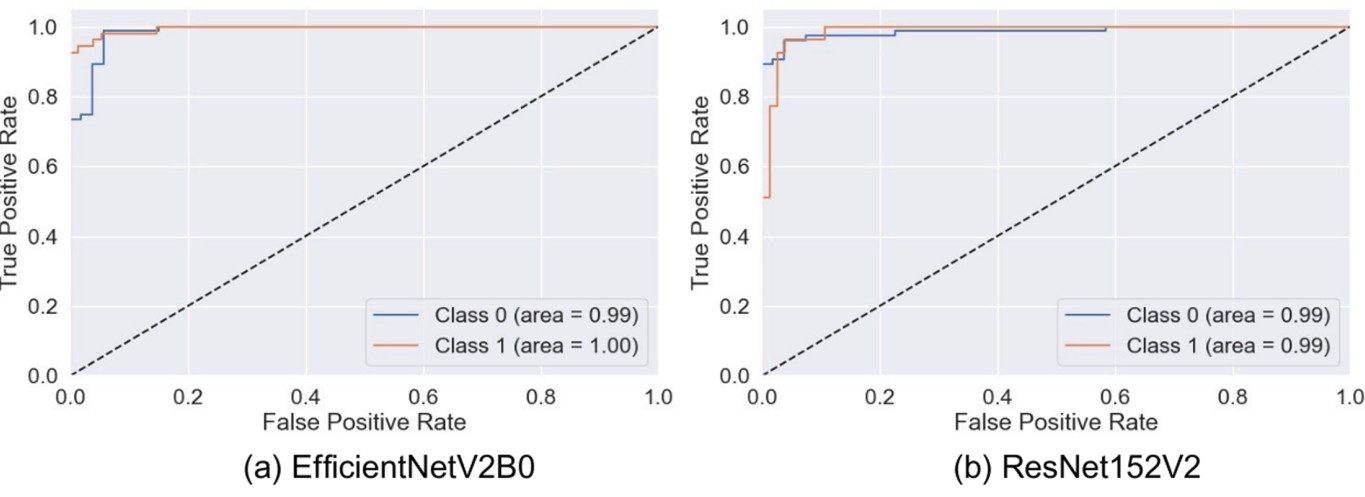

**Fig 9. ROC curves for EfficientNetV2B0 and ResNet152V2.** Here, 0 = infected, 1 = normal.

In most instances, Xception is strong, with one class, there is a slightly lower AUC, which means a marginally greater false positive rate or lower true positive rate.

However, from Fig 10 our proposed model abounds in AUC scores, containing 1.00 in one class and 0.99 in the other, exhibiting an almost perfect classification capability. This indicates that the proposed model, which minimizes trade-offs, has a remarkable capability for discrimination. This makes its ROC curve almost perfect, which will dwell in a high dimension of true positives and yet squeeze the false positives. This would make the model proposed in this study the most appropriate for applications where sensitivity and specificity are important for substantial consequences, such as medical diagnostics. This will result in far more balanced and correct performance in the proposed model with fewer risks of missing true cases or too many false alarms.

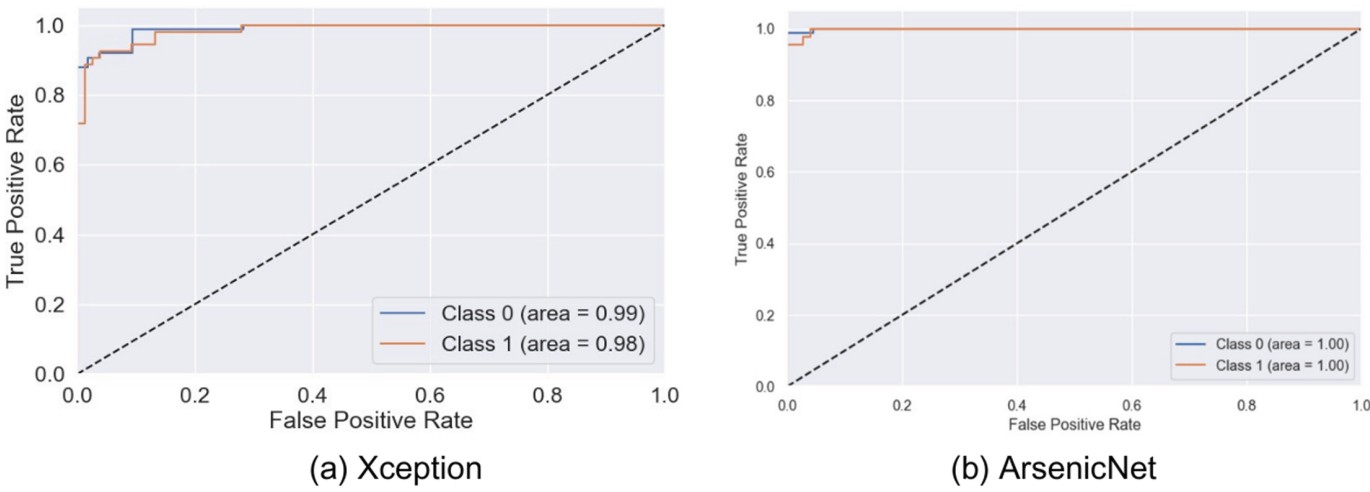

**Fig 10. ROC curves for Xception and Proposed model.** Here, 0 = infected, 1 = normal.

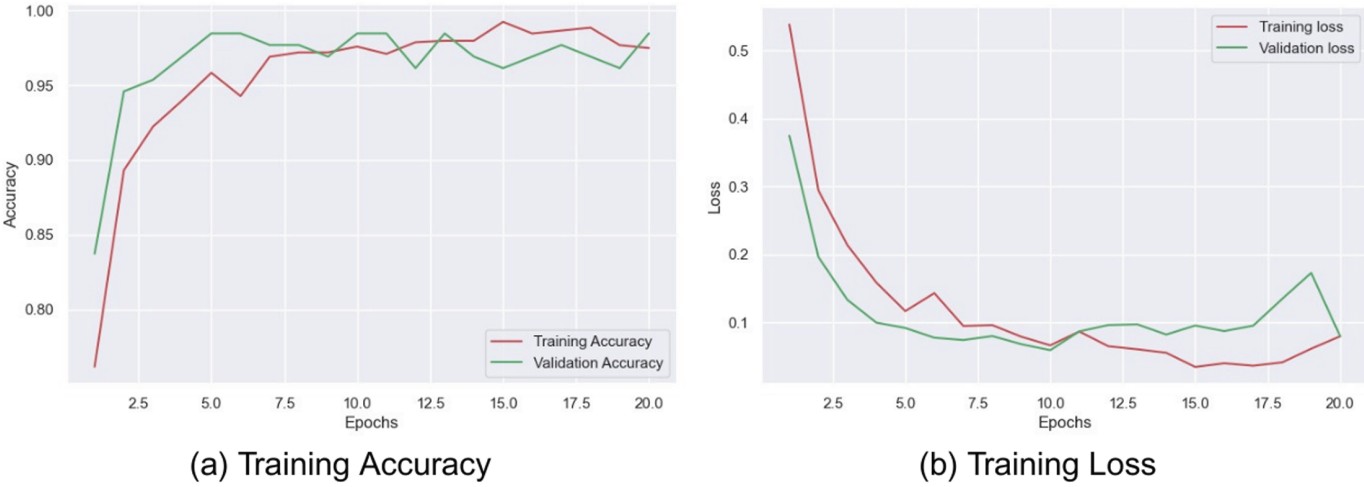

**Fig 11. Proposed model's training performance on 80:10:10 data split: accuracy vs. epochs and loss vs. epochs.**

The following graphs in Fig 11 show the performance of the proposed model for arsenic classification in the training and validation test up to 20 epochs. The accuracy graph shows a steady increase in training and validation accuracies. The training accuracy increases rapidly, reaching perfect levels at a very early stage, more so at the start, and stabilizes. The validation accuracy is also high, slightly fluctuating, but very close to training accuracy, which suggests that the model is learning well and generalizing to test data.

The training and validation losses have significantly decreased during the initial epochs of the loss graph, indicating that the model learns quite quickly. The training loss decreased steadily, while the validation loss attained a low value and maintained it with slight fluctuations in the later epochs. These small fluctuations indicate a minor chance of overfitting, which can result in the model being too tailored to the training data. In summary, the model performed very well and had high accuracy, and a low loss was maintained by overfitting a bit in the last training epochs.

## 4.3 Comparison with state-of-the-art models

In this section, we describe Table 5, which displays the performance of the 7 deep learning models, including the proposed model. Notably, in this scenario, the combination of the Xception and Inception modules, which is our proposed model, exhibits superior

**Table 5. Performance metrics comparison of various models for 80:10:10 data split. Here, 80% for training set, 10% validation set, 10% test set, test accuracy as Test_Acc.**

| Models | Test_Acc | Test_Loss | Sensitivity | Precision | F1 Score |
|---|---|---|---|---|---|
| InceptionV3 [26] | 93.85 | 11.82 | 94.18 | 93.51 | 93.75 |
| VGG19 [27] | 90.77 | 23.77 | 91.76 | 90.77 | 90.71 |
| EfficientNetV2B0 [28] | 93.85 | 17.31 | 93.45 | 93.92 | 93.66 |
| ViT [31] | 86.92 | 24.67 | 86.73 | 86.56 | 86.64 |
| ResNet152V2 [29] | 93.85 | 21.47 | 94.67 | 93.65 | 93.79 |
| Xception [30] | 94.62 | 19.36 | 93.88 | 95.25 | 94.41 |
| Proposed model | 97.69 | 4.92 | 97.52 | 97.76 | 97.63 |

performance and achieves a test accuracy of nearly 97 and a half percent with an impressively low test loss of just under 5, which reflects the robustness and efficiency of our objective. It is noteworthy that the proposed model has achieved the highest F1 score of over 97%, which emphasizes its unique precision and recall balance. It is observable that Xception alone also performs better, with a test accuracy of almost 95% and an F1 score of close to 94 and a half. Besides these, InceptionV3, EfficientNetV2B0, and ResNet152V2 represent equivalent test accuracies of 93.85% , indicating their strong yet slightly lesser performance contrast to the proposed model. However, VGG19 also achieves a moderate recall of almost 92%, suffers from higher test loss at almost 24 and lower test accuracy. Moreover, a closer analysis reveals that the ViT model yields the lowest test accuracy, precision, and F1 score, approximately 87%. These results show that our proposed model outperformed all the single models across all sectors.

The bar chart in Fig 12 shows the training accuracy and loss across different deep learning models, including our proposed model, where 80-10-10 data were split on the dataset for training, testing and validation. Our proposed model, the Xception and Inception module, achieve the highest training accuracy of almost 99 and a half percent, respectively, where the lowest training loss is just under 3, reflecting its efficiency. In contrast, VGG19 also achieved a relatively high training accuracy of almost 94%, where we found a significantly higher training loss of just over 20%, indicating a struggle during training compared to other individual models. Moreover, EfficientNetV2B0 also achieves a higher training accuracy of just passing 98 and a half percent, but there is a low training loss, which is 3.43, similar to InceptionV3. In other models like ResNet152V2, which has a training accuracy of almost 96 percent and a loss of 10.44, Xception has a training accuracy of just past 98 percent and a moderate loss of almost 7, which describes a healthy balance between accuracy and loss. However, for the ViT model, we have the highest training loss among all models, which is approximately 24%, and this model has shown the lowest accuracy rate, which is just over 90%. Overall, the proposed model, a combination of Xception and Inception modules, is underscored and recognized as the most effective model among the individual models.

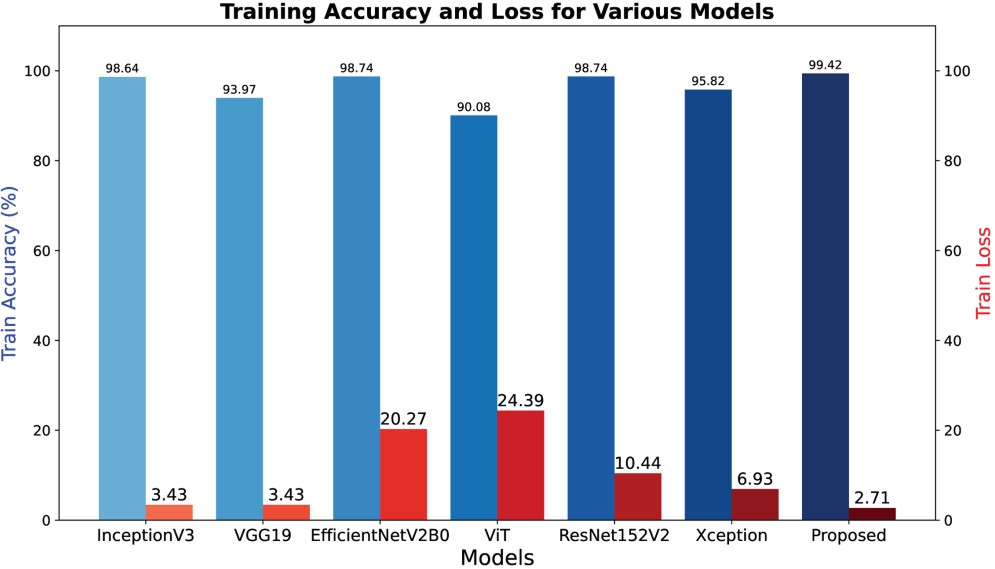

**Fig 12. Training accuracy and loss of the various deep learning models based on 80:10:10 data split.**

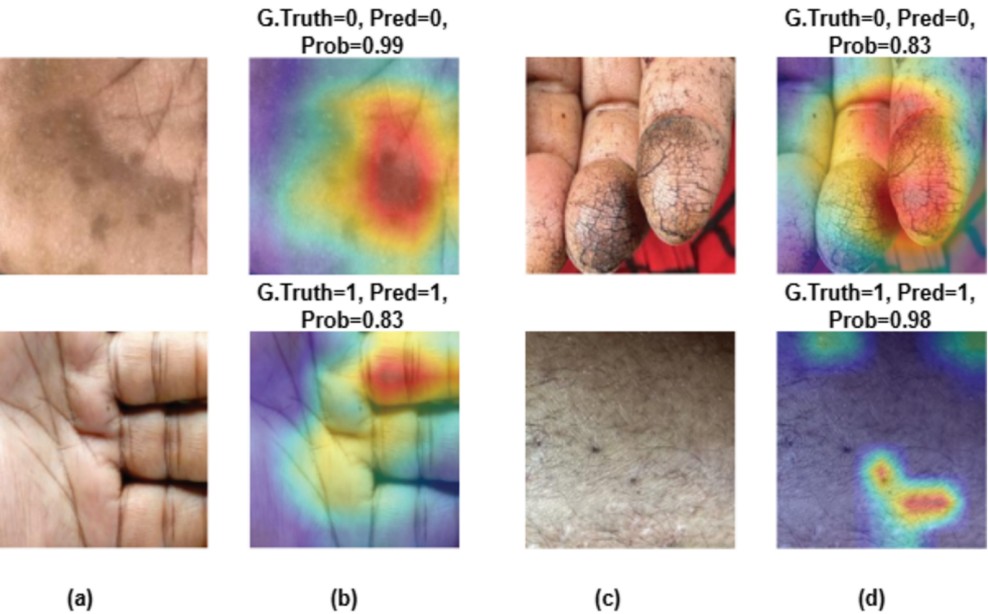

**Fig 13. Grad-CAM visualization (a) and (c) input image, (b) and (d) Grad-CAM overlay on original image. Here, 0=infected, 1=normal.**

## 4.4 Visualization with explainable AI Grad-CAM

Grad-CAM (Gradient-weighted Class Activation Mapping) represents a valuable method to visualize the regions of an image that have the most significant impact on the predictions made by the model. It is a widespread tool for medical image classification to make the model more reliable for clinicians. The heatmap helps us understand how the model makes its decisions, fostering trust in its predictions. The highlighted part of the arsenic skin image had the greatest impact on models's classification decisions. By analysing the heatmap, radiologists can align the model's predictions with their expert knowledge, thereby improving their comprehension and confidence in the findings produced by this system. We applied Grad-CAM to show the model's prediction using a heatmap as shown in Fig 13, Where red, green and yellow colours influenced most to make prediction decisions.

## 5 Conclusion

Arsenic can lead people to death. Worldwide, many people are affected by this deadly disease every year by drinking arsenic-containing water. Early detection and rapid treatment are essential to prevent this and save thousands of lives. Our research highlights a new approach to detecting arsenic skin disease in the human body from image data caused by drinking arsenic water. Our findings demonstrate that the proposed fusion architecture significantly outperforms all other state-of-the-art models by a good margin, with a 3-7% value. Using an image dataset based on Bangladesh, this research can be a vital solution to the critical need that we can use advanced methodology, particularly in rural areas where groundwater is the only primary source of drinking water. Besides, the model's robustness in dealing with the complexity of real-world data focuses on its potential scalable application in many other regions where the arsenic contamination rate is much higher. Furthermore, our research not only focuses on detecting arsenic disease but also provides us with solutions for monitoring

and managing arsenic-related health issues. Introducing a new deep learning model, our research has made a wide path to discover more positive approaches for this health risk problem, which can offer hope for more efficient arsenic detection methods that can be introduced at scale to safeguard public health, particularly in dangerous regions like Bangladesh. Future work will focus on expanding the dataset and will combine this work with other skin diseases in different biological and geographical contexts. Additionally, the semi-supervised methods will also be applied to this dataset. Furthermore, fine-tune the model for better performance in real-world scenarios.

## Author contributions

**Conceptualization:** Md Humaion Kabir Mehedi, Kh. Fardin Zubair Nafis, Krity Haque Charu, Jia Uddin.

**Formal analysis:** Md Humaion Kabir Mehedi, Kh. Fardin Zubair Nafis, M.F. Mridha.

**Investigation:** Krity Haque Charu.

**Methodology:** Md Humaion Kabir Mehedi, Kh. Fardin Zubair Nafis, M.F. Mridha.

**Resources:** Jia Uddin, Md Golam Rabiul Alam.

**Supervision:** M.F. Mridha.

**Validation:** Krity Haque Charu, Jia Uddin, Md Golam Rabiul Alam.

**Visualization:** Md Humaion Kabir Mehedi.

**Writing – original draft:** Md Humaion Kabir Mehedi, Kh. Fardin Zubair Nafis, Krity Haque Charu.

**Writing – review & editing:** Jia Uddin, Md Golam Rabiul Alam, M.F. Mridha.

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
