## [Decision Letter · Decision Letter 0]

19 Nov 2024

PONE-D-24-44589ArsenicNet: An Efficient Way of Arsenic Skin Disease Detection Using Enriched Fusion Xception ModelPLOS ONE

Dear Dr. Mridha,

Thank you for submitting your manuscript to PLOS ONE. After careful consideration, we feel that it has merit but does not fully meet PLOS ONE’s publication criteria as it currently stands. Therefore, we invite you to submit a revised version of the manuscript that addresses the points raised during the review process.

We look forward to receiving your revised manuscript.

Kind regards,

Asadullah Shaikh, Ph.D.

Academic Editor

PLOS ONE

Journal Requirements:

Reviewers' comments:

Reviewer's Responses to Questions

**Comments to the Author**

1. Is the manuscript technically sound, and do the data support the conclusions?

Reviewer #1: Partly

Reviewer #2: Partly

Reviewer #3: Partly

2. Has the statistical analysis been performed appropriately and rigorously? 

Reviewer #1: Yes

Reviewer #2: Yes

Reviewer #3: Yes

3. Have the authors made all data underlying the findings in their manuscript fully available?

Reviewer #1: Yes

Reviewer #2: Yes

Reviewer #3: Yes

4. Is the manuscript presented in an intelligible fashion and written in standard English?

Reviewer #1: No

Reviewer #2: No

Reviewer #3: Yes

5. Review Comments to the Author

Reviewer #1: The manuscript introduces "ArsenicNet," a novel fusion model for arsenic-induced skin disease detection that combines the Xception model with an Inception module. The paper aims to address a significant public health issue by providing an accurate and efficient approach to detect arsenic contamination-related skin conditions, particularly for populations in Bangladesh who rely heavily on groundwater. With an impressive accuracy of 97.69%, the model outperformed five state-of-the-art deep learning models. Using the "ArsenicSkinImageBD" dataset, the authors argue that ArsenicNet is a promising tool for public health diagnostics.

Major Comments:

1. Model Justification and Architecture:

Comment: The integration of Xception with an Inception module is creative, yet the manuscript lacks a detailed rationale for choosing this specific combination. While the authors provide a block diagram and description, additional explanation of how each module contributes to performance and why they were chosen over other architectures is necessary.

Recommendation: Include a more thorough discussion on why the Xception-Inception fusion is preferred, specifically highlighting the roles of multi-scale feature extraction and depthwise separable convolutions in improving accuracy for arsenic skin disease detection.

2. Dataset Limitations and Generalizability:

Comment: The dataset used in this study, "ArsenicSkinImageBD," is highly specific to Bangladesh. While appropriate for this study, this limited scope risks reducing the model's generalizability across broader populations and environments. Arsenic contamination affects regions beyond Bangladesh, and the model might face performance drops in other contexts due to demographic and environmental variations in skin disease manifestation.

Recommendation: Address this limitation by discussing potential generalizability issues and the need for future external validation. If feasible, suggest synthetic augmentation techniques or access to additional datasets for a more comprehensive evaluation.

3. Lack of Explainability in Model Predictions:

Comment: In a clinical setting, explainability is crucial. The paper currently lacks any application of Explainable AI (XAI) methods to enhance the interpretability of the model’s predictions, especially given the model’s sensitivity to complex patterns in skin images.

Recommendation: Incorporate XAI tools, such as SHAP or Grad-CAM, to illustrate how the model makes decisions. Visual representations of feature importance or heatmaps over skin images could be particularly beneficial in understanding the model’s decision-making process and would enhance trust in clinical applications.

4. Comparison with Modern Techniques:

Comment: While the model achieves significant accuracy, the comparative analysis is limited to relatively established deep learning architectures like InceptionV3, VGG19, and ResNet152V2. Given the rapid advancements in computer vision, including the adoption of Vision Transformers (ViT) and ensemble learning methods, the benchmarks used in this study might not fully demonstrate the model's relative performance.

Recommendation: Test the model against Vision Transformers, ensemble learning techniques, or hybrid models that have recently shown promising results in image classification tasks. This would provide a more comprehensive assessment and enhance the study's relevance.

5. Performance Metrics and Thresholds:

Comment: The paper reports high accuracy, but it does not provide adequate detail on other critical metrics, such as sensitivity, specificity, and F1-scores, across different thresholds. These metrics are especially relevant in medical applications, where minimizing false negatives (missed disease cases) is often more critical than maximizing accuracy alone.

Recommendation: Expand the performance analysis to include a discussion of sensitivity, specificity, and the implications of various classification thresholds. An ROC curve and AUC values provide an initial sense of balance, but concrete metrics on false negatives and false positives at different thresholds are necessary to evaluate the model's robustness and clinical applicability.

6. Reproducibility and Model Training Details:

Comment: Although the manuscript includes some hyperparameter settings, it lacks comprehensive details on model training, such as batch size, data preprocessing specifics, data augmentation techniques, and hardware configurations. These are essential for replicating the study and ensuring its robustness.

Recommendation: Add a dedicated section that details all preprocessing steps, including data normalization, resizing, augmentation parameters, and specific hardware configurations used during training. Provide detailed values for all hyperparameters, learning rates, and decay schedules to facilitate reproducibility.

7. Broader Implications and Ethical Considerations:

Comment: The health implications of arsenic detection are significant, and while the study highlights the potential of ArsenicNet for public health, it lacks a broader discussion on ethical considerations, such as privacy concerns, data security, and the implications of false positives and negatives in a medical setting.

Recommendation: Address the ethical considerations of deploying such models in healthcare, including privacy safeguards, the potential impacts of misdiagnosis, and plans to manage data confidentiality in clinical applications.

Minor Comments:

Abstract: The abstract should briefly highlight the limitations of the dataset and the need for broader validation.

Figures: Figures related to the model architecture could be made clearer with improved annotations on each component.

Terminology Consistency: Ensure consistency in terminology, especially regarding technical terms like "false positives," "false negatives," "precision," and "sensitivity."

Reviewer #2: The paper presents an interesting topic. However, it needs general language and structural editing to enhance its readability.

For instance, in the related work, a statement such as "In this paper, authors tried to get information of measuring arsenic in groundwater [5]. Authors wanted to draw a concern regarding..." is a bit low for this kind of research paper.

I recommend proper language editing to enhance the paper's readability and resubmission for proper review.

Reviewer #3: 1.Ensure a logical flow from data preparation, model architecture, hyperparameter tuning, experimental setup, and result analysis. This will help readers follow your process easily.

2.Provide clear justifications for why Xception and an inception module were combined and how this particular choice impacts classification.

3.Clearly justify the choice of data split (80-10-10 vs. 70-15-15). Discuss how this choice impacts model generalization and robustness, ideally with cross-validation results or a graph showing performance for each split.

4.Provide more in-depth information on the inception module, explaining how each filter size contributes to multi-scale feature extraction and its impact on model performance.

5.Provide more context around why certain models perform better on this dataset. If available, show how similar architectures performed on similar datasets.

6.Include a more in-depth analysis of confusion matrix results and the ROC curves for each model. Highlight why certain models had higher false positives or false negatives.

7.Use bar graphs or spider charts to compare models across multiple metrics (accuracy, F1, precision, recall). This will offer a clearer view of each model’s strengths and weaknesses.

8. Acknowledge any limitations, such as dataset size, imbalance, or possible overfitting issues, and discuss how these could affect the model's performance in real-world scenarios.

9.Propose specific future work directions, like exploring other architectures, optimizing additional hyperparameters, or testing on larger datasets.

6. PLOS authors have the option to publish the peer review history of their article (what does this mean?). If published, this will include your full peer review and any attached files.

Reviewer #1: No

Reviewer #2: No

Reviewer #3: No

---

## [Author Response · Author response to Decision Letter 1]

26 Jan 2025

Reviewer#1:

The manuscript introduces "ArsenicNet," a novel fusion model for arsenic-induced skin disease detection that combines the Xception model with an Inception module. The paper aims to address a significant public health issue by providing an accurate and efficient approach to detect arsenic contamination-related skin conditions, particularly for populations in Bangladesh who rely heavily on groundwater. With an impressive accuracy of 97.69%, the model outperformed five state-of-the-art deep learning models. Using the "ArsenicSkinImageBD" dataset, the authors argue that ArsenicNet is a promising tool for public health diagnostics.

Reviewer#1, Concern # 1 (please list here): Model Justification and Architecture:

Comment: The integration of Xception with an Inception module is creative, yet the manuscript lacks a detailed rationale for choosing this specific combination. While the authors provide a block diagram and description, additional explanation of how each module contributes to performance and why they were chosen over other architectures is necessary.

Recommendation: Include a more thorough discussion on why the Xception-Inception fusion is preferred, specifically highlighting the roles of multi-scale feature extraction and depth wise separable convolutions in improving accuracy for arsenic skin disease detection.

Author response: Thanks for the comment.

Author action: Xception uses depth wise separable convolutions to extract strong spatial features uniformly across the feature map where a multi-scale processing module such as Inception helps the model learn features of different sizes and complexity. The fine-grained features of Xception are enhanced by the Inception module, making the model more adaptable to different datasets. The modular design of Inception and Xception's computational efficiency makes the architecture lightweight. The trainable parameters for Xception were 51,512,578 whereas the proposed combination had 814,962. The fixed Xception architecture can be made flexible by changing the filter sizes or branch configurations in the Inception module to suit the dataset. Medical images, such as skin disease images, benefit from this combination because the Xception backbone extracts strong features, and the Inception module captures context and scale-specific information. Without the Inception module the Xception test accuracy is 94.62% and with the Inception module 97.69%.

The relevant text has been added in Section 3.4, page no 8 and all the revised text is highlighted in yellow colour.________________________________________

Reviewer#1, Concern # 2 (please list here): Dataset Limitations and Generalizability:

Comment: The dataset used in this study, "ArsenicSkinImageBD," is highly specific to Bangladesh. While appropriate for this study, this limited scope risks reducing the model's generalizability across broader populations and environments. Arsenic contamination affects regions beyond Bangladesh, and the model might face performance drops in other contexts due to demographic and environmental variations in skin disease manifestation.

Recommendation: Address this limitation by discussing potential generalizability issues and the need for future external validation. If feasible, suggest synthetic augmentation techniques or access to additional datasets for a more comprehensive evaluation.

Author response: Thank you for your concern.

Author action: We used “ArsenicSkinImageBD” dataset for our study. The dataset sample has been collected from 4 different villages in Bangladesh. We did not find any other public arsenic skin image based dataset. So we could not do any external validation, but in the future we will try to incorporate other skin disease datasets with our used dataset in this study. We believe our proposed model may not drop performance in other contexts due to demographic and environmental variations in skin disease manifestation.

Reviewer#1, Concern # 3(please list here): Lack of Explainability in Model Predictions:

Comment: In a clinical setting, explainability is crucial. The paper currently lacks any application of Explainable AI (XAI) methods to enhance the interpretability of the model’s predictions, especially given the model’s sensitivity to complex patterns in skin images.

Recommendation: Incorporate XAI tools, such as SHAP or Grad-CAM, to illustrate how the model makes decisions. Visual representations of feature importance or heatmaps over skin images could be particularly beneficial in understanding the model’s decision-making process and would enhance trust in clinical applications.

Author response: Thanks for the observation.

Author action: We applied Grad-Cam to understand the decision-making process of the model and would enhance trust in clinical applications. Section “4.4 Visualization with Explainable AI Grad-CAM” added with detailed explanations and examples. The revised text has been highlighted with in yellow color on page no 17.

Reviewer#1, Concern # 4 (please list here): Comparison with Modern Techniques:

Comment: While the model achieves significant accuracy, the comparative analysis is limited to relatively established deep learning architectures like InceptionV3, VGG19, and ResNet152V2. Given the rapid advancements in computer vision, including the adoption of Vision Transformers (ViT) and ensemble learning methods, the benchmarks used in this study might not fully demonstrate the model's relative performance.

Recommendation: Test the model against Vision Transformers, ensemble learning techniques, or hybrid models that have recently shown promising results in image classification tasks. This would provide a more comprehensive assessment and enhance the study's relevance.

Author response: Thank you for the comment.

Author action: As we already know, the transformer model requires a large amount of data for training, whereas our dataset contains only 1287 samples. We have tried different transformer models such as ViT, Swin and Deit. However, without ViT, none could perform significantly well. The ViT results are reported in Table 5

Reviewer#1, Concern # 5 (please list here): Performance Metrics and Thresholds:

Comment: The paper reports high accuracy, but it does not provide adequate detail on other critical metrics, such as sensitivity, specificity, and F1-scores, across different thresholds. These metrics are especially relevant in medical applications, where minimizing false negatives (missed disease cases) is often more critical than maximizing accuracy alone.

Recommendation: Expand the performance analysis to include a discussion of sensitivity, specificity, and the implications of various classification thresholds. An ROC curve and AUC values provide an initial sense of balance, but concrete metrics on false negatives and false positives at different thresholds are necessary to evaluate the model's robustness and clinical applicability.

Author response: Thank you for the comment.

Author action: Our model not only achieved higher accuracy but also a higher recall/sensitivity score, which is very important in medical applications. In Table 5, all performance metrics are listed.

Reviewer#1, Concern # 6 (please list here): Reproducibility and Model Training Details:

Comment: Although the manuscript includes some hyperparameter settings, it lacks comprehensive details on model training, such as batch size, data preprocessing specifics, data augmentation techniques, and hardware configurations. These are essential for replicating the study and ensuring its robustness.

Recommendation: Add a dedicated section that details all preprocessing steps, including data normalization, resizing, augmentation parameters, and specific hardware configurations used during training. Provide detailed values for all hyperparameters, learning rates, and decay schedules to facilitate reproducibility.

Author response: Thank you for the suggestion.

Author action: We have already provided all the data preprocessing steps, including data normalization, resizing, and augmentation parameters, in Section 3.2. The hardware configurations are listed in Table 4. Moreover, Table 3 provides all hyperparameters.

Reviewer#1, Concern # 7 (please list here): Broader Implications and Ethical Considerations:

Comment: The health implications of arsenic detection are significant, and while the study highlights the potential of ArsenicNet for public health, it lacks a broader discussion on ethical considerations, such as privacy concerns, data security, and the implications of false positives and negatives in a medical setting.

Recommendation: Address the ethical considerations of deploying such models in healthcare, including privacy safeguards, the potential impacts of misdiagnosis, and plans to manage data confidentiality in clinical applications.

Author response: Thank you for the comment.

Author action: We used a publicly available dataset hosted in Mendeley Data for our study. In addition, the dataset is anonymous, so there is no chance that the patient information will be licked. The authors of the dataset maintained all ethical considerations and privacy of the patient.

Reviewer#1, Concern # 8 (please list here): Abstract: The abstract should briefly highlight the limitations of the dataset and the need for broader validation.

Author response: Thank you for your comment.

Author action: We have added dataset limitations in the abstract.

Reviewer#1, Concern # 9 (please list here): Figures: Figures related to the model architecture could be made clearer with improved annotations on each component.

Author response: Thank you for your concern.

Author action: We have updated Figures 5, 6 and 7.

Reviewer#1, Concern # 10 (please list here): Terminology Consistency: Ensure consistency in terminology, especially regarding technical terms like "false positives," "false negatives," "precision," and "sensitivity."

Author response: Thank you for the feedback.

Author action: We have updated the manuscript.

Reviewer#2: The paper presents an interesting topic. However, it needs general language and structural editing to enhance its readability.

For instance, in the related work, a statement such as "In this paper, authors tried to get information of measuring arsenic in groundwater [5]. Authors wanted to draw a concern regarding..." is a bit low for this kind of research paper.

Reviewer#2, Concern # 1 (please list here): I recommend proper language editing to enhance the paper's readability and resubmission for proper review.

Author response: Thank you for the comment.

Author action: We have revised the manuscript and proofread by a native speaker.

Reviewer#3, Concern #1(please list here): Ensure a logical flow from data preparation, model architecture, hyperparameter tuning, experimental setup, and result analysis. This will help readers follow your process easily.

Author response: Thank you for the suggestion.

Author action: We have updated according to your suggestions.________________________________________

Reviewer#3, Concern #2(please list here): Provide clear justifications for why Xception and an inception module were combined and how this particular choice impacts classification

Author response: Thank you for the feedback.

Author action: Xception uses depth wise separable convolutions to extract strong spatial features uniformly across the feature map where a multi-scale processing module such as Inception helps the model learn features of different sizes and complexity. The fine-grained features of Xception are enhanced by the Inception module, making the model more adaptable to different datasets. The modular design of Inception and Xception's computational efficiency make the architecture lightweight. The trainable parameters for Xception were 51,512,578 whereas the proposed combination had 814,962. The fixed Xception architecture can be made flexible by changing the filter sizes or branch configurations in the Inception module to suit the dataset. Medical images, such as skin disease images, benefit from this combination because the Xception backbone extracts strong features, and the Inception module captures context and scale-specific information.

The relevant text are added in Section 3.4, page no 8 and all the revised text is highlighted in yellow colour.

Reviewer#3, Concern #3(please list here): Clearly justify the choice of data split (80-10-10 vs. 70-15-15). Discuss how this choice impacts model generalization and robustness, ideally with cross-validation results or a graph showing performance for each split.

Author response: Thank you for the observation.

Author action: For our study, we used different data splits, such as 80-10-10 vs. 70-15-15 but obtained the best performance on 80-10-10.

Reviewer#3, Concern #4(please list here): Provide more in-depth information on the inception module, explaining how each filter size contributes to multi-scale feature extraction and its impact on model performance.

Author response: Thank you for the comment.

Author action: Xception uses depth wise separable convolutions to extract strong spatial features uniformly across the feature map where a multi-scale processing module such as Inception helps the model learn features of different sizes and complexity. The fine-grained features of Xception are enhanced by the Inception module, making the model more adaptable to different datasets. The modular design of Inception and Xception's computational efficiency make the architecture lightweight. The trainable parameters for Xception were 51,512,578 whereas the proposed combination had 814,962. The fixed Xception architecture can be made flexible by changing the filter sizes or branch configurations in the Inception module to suit the dataset. Medical images, such as skin disease images, benefit from this combination because the Xception backbone extracts strong features, and the Inception module captures context and scale-specific information. We have tried with different filter size but got the best outcome with the proposed set given in Fig. 4. Equations 2, 3, 4, 5 and 6 are related to filter and kernels of Inception module.

Reviewer#3, Concern #5(please list here): Provide more context around why certain models perform better on this dataset. If available, show how similar architectures are performed on similar datasets.

Author response: Thank you for the concern.

Author action:

Reviewer#3, Concern #6(please list here): Include a more in-depth analysis of confusion matrix results and the ROC curves for each model. Highlight why certain models had higher false positives or false negatives.

Author response: Thank you for the remarks.

Author action: We have already given it in Section 4.2.

Reviewer#3, Concern #7(please list here): Use bar graphs or spider charts to compare models across multiple metrics (accuracy, F1, precision, recall). This will offer a clearer view of each model’s strengths and weaknesses

Author response: Thank you for the suggestion.

Author action: We have shown multiple metrics (accuracy, precision, recall/ sensitivity and F1 score) in Table 5.

Reviewer#3, Concern #8(please list here): Acknowledge any limitations, such as dataset size, imbalance, or possible overfitting issues, and discuss how these could affect the model's performance in real-world scenarios.

Author response: Thank you for the concern.

Author action: Dataset size with only 1287 samples, is one of the main limitations. As the dataset is small, there is a chance of model overfitting. Therefore, we addressed these issues using early callback function. If we could train on a large dataset, the performance of the model would increase, which would increase its usability in real-world scenarios.

Reviewer#3, Concern #9(please list here): Propose specific future work directions, like exploring other architectures, optimizing additional hyperparameters, or testing on larger datasets.

Author response: Thank you for the concern.

Author action: Future work will focus on expanding the dataset and will combine this work with other skin diseases in different biological and geographical con

---

## [Decision Letter · Decision Letter 1]

18 Feb 2025

PONE-D-24-44589R1ArsenicNet: An Efficient Way of Arsenic Skin Disease Detection Using Enriched Fusion Xception ModelPLOS ONE

Dear Dr. Mridha,

Thank you for submitting your manuscript to PLOS ONE. After careful consideration, we feel that it has merit but does not fully meet PLOS ONE’s publication criteria as it currently stands. Therefore, we invite you to submit a revised version of the manuscript that addresses the points raised during the review process.

We look forward to receiving your revised manuscript.

Kind regards,

Asadullah Shaikh, Ph.D.

Academic Editor

PLOS ONE

Journal Requirements:

Reviewers' comments:

Reviewer's Responses to Questions

**Comments to the Author**

1. If the authors have adequately addressed your comments raised in a previous round of review and you feel that this manuscript is now acceptable for publication, you may indicate that here to bypass the “Comments to the Author” section, enter your conflict of interest statement in the “Confidential to Editor” section, and submit your "Accept" recommendation.

Reviewer #2: All comments have been addressed

Reviewer #3: All comments have been addressed

2. Is the manuscript technically sound, and do the data support the conclusions?

Reviewer #2: Partly

Reviewer #3: Yes

3. Has the statistical analysis been performed appropriately and rigorously? 

Reviewer #2: (No Response)

Reviewer #3: Yes

4. Have the authors made all data underlying the findings in their manuscript fully available?

Reviewer #2: Yes

Reviewer #3: Yes

5. Is the manuscript presented in an intelligible fashion and written in standard English?

Reviewer #2: Yes

Reviewer #3: Yes

6. Review Comments to the Author

Reviewer #2: I commend the authors for their effort in improving the manuscript based on the previous comments, but I still believe there are still a few things to address.

Additional Comments:

1. Language and Structure: Although the authors indicated that the manuscript has been proofread, some sections still require general language and structural editing for improved readability. For example, in the abstract, a statement like "The proposed model achieved 97.69% accuracy, along with 97.63% F1 score and reached the top"; in section 4.3, a statement like "More over, if we notice closely, then we can see that we have got the lowest test accuracy,..."; etc.

Recommendation: Professional language editing will perhaps be much better.

2. The figure illustrating the model architecture needs clarity. Adequate annotations explaining each component and its role in the architecture are important.

3. Dataset size: The author acknowledged that the dataset is small. This is a big concern as to the generalisability of the model findings.

Recommendation: The author should include a comprehensive discussion (also in the abstract) on the limitations of the research (particularly the chances of model overfitting).

Reviewer #3: I recommend to accept the paper for publishing since all my comments were addressed, the modifications has enhanced the paper quality.

7. PLOS authors have the option to publish the peer review history of their article (what does this mean?). If published, this will include your full peer review and any attached files.

Reviewer #2: No

Reviewer #3: No

---

## [Author Response · Author response to Decision Letter 2]

3 Mar 2025

Reviewer#2: I commend the authors for their effort in improving the manuscript based on the previous comments, but I still believe there are still a few things to address.

Reviewer#2, Concern # 1 (please list here): Language and Structure: Although the authors indicated that the manuscript has been proofread, some sections still require general language and structural editing for improved readability. For example, in the abstract, a statement like "The proposed model achieved 97.69% accuracy, along with 97.63% F1 score and reached the top"; in section 4.3, a statement like "More over, if we notice closely, then we can see that we have got the lowest test accuracy,..."; etc.

Recommendation: Professional language editing will perhaps be much better.

Author response: Thank you for the comment.

Author action: We have revised the manuscript and proofread it by a native speaker.

Reviewer#2, Concern #2(please list here): The figure illustrating the model architecture needs clarity. Adequate annotations explaining each component and its role in the architecture are important.

Author response: Thank you for the suggestion.

Author action: We have clearly drawn and annotated the model architecture figure. We have also provided a detailed description of each component's role.

Reviewer#2, Concern #3(please list here): Dataset size: The author acknowledged that the dataset is small. This is a big concern as to the generalisability of the model findings.

Recommendation: The author should include a comprehensive discussion (also in the abstract) on the limitations of the research (particularly the chances of model overfitting).

Author response: Thank you for the feedback.

Author action: As we already said, the dataset is small in size and there are no alternative datasets available. So, there might be a chance of model overfitting. To prevent this issue, we applied different data augmentation techniques, such as horizontal flip, width, and height shift. In addition, we used early stopping function to prevent overfitting issues by monitoring val_loss.

---

## [Decision Letter · Decision Letter 2]

21 Mar 2025

ArsenicNet: An Efficient Way of Arsenic Skin Disease Detection Using Enriched Fusion Xception Model

PONE-D-24-44589R2

Dear Dr. Mridha,

We’re pleased to inform you that your manuscript has been judged scientifically suitable for publication and will be formally accepted for publication once it meets all outstanding technical requirements.

Kind regards,

Asadullah Shaikh, Ph.D.

Academic Editor

PLOS ONE

Additional Editor Comments (optional):

Reviewers' comments:

Reviewer's Responses to Questions

**Comments to the Author**

1. If the authors have adequately addressed your comments raised in a previous round of review and you feel that this manuscript is now acceptable for publication, you may indicate that here to bypass the “Comments to the Author” section, enter your conflict of interest statement in the “Confidential to Editor” section, and submit your "Accept" recommendation.

Reviewer #2: All comments have been addressed

2. Is the manuscript technically sound, and do the data support the conclusions?

Reviewer #2: Yes

3. Has the statistical analysis been performed appropriately and rigorously? 

Reviewer #2: N/A

4. Have the authors made all data underlying the findings in their manuscript fully available?

Reviewer #2: Yes

5. Is the manuscript presented in an intelligible fashion and written in standard English?

Reviewer #2: Yes

6. Review Comments to the Author

Reviewer #2: The authors have been able to address all comments raised in the previous review. No further comments.

7. PLOS authors have the option to publish the peer review history of their article (what does this mean?). If published, this will include your full peer review and any attached files.

Reviewer #2: No

---

## [Editor Report · Acceptance letter]

PONE-D-24-44589R2

PLOS ONE

Dear Dr. Mridha,

I'm pleased to inform you that your manuscript has been deemed suitable for publication in PLOS ONE. Congratulations! Your manuscript is now being handed over to our production team.

Kind regards,

on behalf of

Prof. Asadullah Shaikh

Academic Editor

PLOS ONE